# Dynamic Nuclear Polarization of Biomembrane Assemblies

**DOI:** 10.3390/biom10091246

**Published:** 2020-08-27

**Authors:** Nhi T. Tran, Frédéric Mentink-Vigier, Joanna R. Long

**Affiliations:** 1Department of Chemistry, University of Florida, Gainesville, FL 32611, USA; nhitran@mit.edu; 2National High Magnetic Field Laboratory, Florida State University, Tallahassee, FL 32310, USA; fmentink@magnet.fsu.edu; 3Department of Biochemistry & Molecular Biology and McKnight Brain Institute, University of Florida, Gainesville, FL 32610, USA

**Keywords:** Dynamic Nuclear Polarization (DNP), Nuclear Magnetic Resonance (NMR), solid-state nuclear magnetic resonance (ssNMR), membrane proteins, membrane active peptides

## Abstract

While atomic scale structural and dynamic information are hallmarks of nuclear magnetic resonance (NMR) methodologies, sensitivity is a fundamental limitation in NMR studies. Fully exploiting NMR capabilities to study membrane proteins is further hampered by their dilution within biological membranes. Recent developments in dynamic nuclear polarization (DNP), which can transfer the relatively high polarization of unpaired electrons to nuclear spins, show promise for overcoming the sensitivity bottleneck and enabling NMR characterization of membrane proteins under native-like conditions. Here we discuss fundamental aspects of DNP-enhanced solid-state NMR spectroscopy, experimental details relevant to the study of lipid assemblies and incorporated proteins, and sensitivity gains which can be realized in biomembrane-based samples. We also present unique insights which can be gained from DNP measurements and prospects for further development of the technique for elucidating structures and orientations of membrane proteins in native lipid environments.

## 1. Introduction

Solid-state NMR spectroscopy (ssNMR) is distinctly capable of determining membrane protein structure and dynamics at high resolution within native lipid environments [1,2,3,4]; it can also provide unique insights into lipid organization and dynamics [5,6]. However, the application range of NMR spectroscopy is sensitivity restricted due to the small transition energies involved resulting in low detectable polarization. In this context, membrane protein structural biology via ssNMR spectroscopy is particularly challenging due to protein dilution within lipid membranes, the smaller protein yields typically observed compared to soluble proteins, and the structural heterogeneity often observed even under optimal sample conditions further restricting the sensitivity. These obstacles lead to NMR signal averaging that substantially increases experimental measurement times and, in many cases, negates progress due to protein aggregation and/or sample breakdown over the time course of a long experiment. In addition to the inherent insensitivity, for protein samples with structural heterogeneity, broad and overlapped NMR resonances preclude NMR chemical shift assignments. Even for structurally homogeneous samples, proteins larger than 10 kDa typically require specific isotope labeling and/or isotope dilution strategies to achieve sufficient resolution for chemical shift assignments, increasing the number of samples which must be made and characterized. Together, the substantial time and financial commitments for successfully characterizing membrane proteins in their native lipid environments using ssNMR are frequently insurmountable barriers limiting the wide-scale adaption of ssNMR spectroscopy.

To increase signal strength, dynamic nuclear polarization (DNP) can transfer the relatively high polarization of unpaired electrons to nuclear spins (Figure 1A), theoretically improving signal to noise by a factor of 660. Magic angle spinning DNP (MAS-DNP) NMR has made large strides in improving the sensitivity of biomolecular ssNMR experiments and illuminating answers to important biological mechanisms not amenable to traditional ssNMR methods [7,8,9,10,11,12,13]. Current efforts in MAS-DNP NMR are being invested in the design and synthesis of efficient polarizing agents (PAs) for high field NMR applications [14,15,16,17,18,19] and in engineering stable microwave sources and NMR probes that can uniformly deliver microwaves into NMR samples [20,21,22,23,24]. Recent mechanistic studies characterizing polarization transfer pathways within large spin systems under MAS-DNP conditions have provided insights into how further gains might be realized [25,26]. Rapid progress in these areas have widely benefited biological applications of MAS-DNP-enhanced ssNMR spectroscopy. As improvements in DNP hardware enable its adoption at higher magnetic fields with faster MAS, it is anticipated gains in resolution will further its utility in characterizing complex biological systems, particularly if PA development for high field applications keeps pace with technologies breakthroughs.

In addition to advances in instrumentation and PA design, careful attention to sample preparation strategies for membrane proteins is necessary for optimal DNP enhancement and sensitivity gains. In this work we systematically discuss the sample and instrumentation requirements for MAS-DNP ssNMR experiments and specifically examine how maximal DNP signal enhancements can be achieved in biomembrane samples while preserving biologically relevant conditions. Current MAS-DNP ssNMR approaches can demonstrably enhance NMR signals by one-to-two orders of magnitude relative to conventional ssNMR spectroscopy. This enables the use of much smaller protein quantities, substantially reducing the duration and cost of sample production. Additionally, the increased sensitivity allows for protein dilution to biologically relevant concentrations within lipid membrane preparations (e.g., >50 lipid molecules per protein). Recent work showing the successful DNP enhancement of NMR signals within intact cells [27,28,29,30] suggests that ultimately membrane proteins could be studied using cell membrane isolates, bypassing extraction and solubilization protocols that have limited their viability. Additionally, signal enhancements can be utilized to study more challenging NMR-active nuclei such as ^17^O [31] and ^43^Ca [32,33], providing novel insights into membrane protein mechanisms.

## 2. DNP-Enhanced Solid-State NMR Spectroscopy

### 2.1. Basic Requirements

Due to its effectiveness, the cross-effect (CE) is widely used for MAS-DNP at high magnetic field strengths [17,18,34,35]. The CE effect occurs when two dipolar coupled electrons, coupled to surrounding nuclei, meet the condition that their respective Larmor frequency difference matches the nuclear Larmor frequency. Depending on the electron polarization difference with respect to the nuclei’s polarization there can be an increase or a decrease of the nuclear polarization with DNP. By irradiating the electron spectrum (EPR) with an appropriate MW irradiation frequency, a significant electron polarization difference is generated that leads to an increase of the polarization for nuclei surrounding the biradical pair via the CE [36,37]. In general, the CE is obtained using wide line EPR biradicals, such as nitroxides, that have a large g-anisotropy allowing matching of the CE condition at high magnetic fields and efficient polarization transfer. Under MAS, the CE mechanism obeys the same principles but is more complex due to the time dependence induced by rotation of the sample which modulates the interactions. First described independently by Thurber et al. and Mentink-Vigier et al. [34,35] the time dependent behavior of {e_a_-e_b_-n} energy levels allows for state mixing over the course of a rotor period enabling energy level anti-crossings dubbed “rotor events” [38]. There are three main rotor events that lead to efficient buildup of nuclear spin polarization under MAS: (1) the microwave (MW) rotor event, (2) the electron dipolar/exchange interaction (D/J) rotor event, and (3) the cross-effect (CE) rotor event. During a MW rotor event, one of the electron Larmor frequencies matches the microwave frequency, ω_a/b_ = ω_MW_, which induces a polarization difference between the coupled electrons. High power microwave irradiation helps generate this polarization difference. The second type of rotor event (D/J) tends to exchange the polarization difference between the two electrons when their respective Larmor frequencies are equal, ω_a_ = ω_b_. The exchange is adiabatic if the dipolar coupling (D) and exchange interaction (J) are strong enough vis a vis the rate of the crossing [34,39]. In this case, the absolute electron polarization difference is maintained; otherwise it is decreased. This polarization difference is essential for the CE mechanism as the CE rotor event (3) adiabatically exchanges with the nuclear polarization when |ω_a_-ω_b_|~ω_n_. Thus, a large electron polarization difference is likely to enhance the nuclear polarization. The efficiency of this polarization transfer is primarily dictated by the dipolar/exchange interaction strength, the hyperfine coupling and the magnetic field [34,35,37]. As a consequence, a strong dipolar/exchange interaction generates faster nuclear polarization build up, while a higher magnetic field slows it [26]. Relaxation also has a role in electron to nuclear polarization transfer: with short electron relaxation times, the electrons’ polarization difference generated by the MW rotor event is not maintained for the time needed to adiabatically exchange with the nuclear polarization [26,35]. Similarly, a short nuclear relaxation time prevents a large nuclear polarization from being reached [37,38,39,40]. Thus, more efficient polarization is enabled by lowering temperature. which lengthens both electron and nuclear relaxation times [41,42,43]. Finally, it is important to stress that this description is simplified for pedagogical reasons. For more details, the reader is referred to recent reviews [44,45,46].

#### 2.1.1. Instrumentation

Commercial implementations of DNP rely on cryogenic cooling via liquid nitrogen. Stable spinning can be achieved down to ~90–100 K with moderate MAS frequencies (10–40 kHz, depending on the size of the sample rotor). Continuous wave irradiation is provided by either a gyrotron [22,47,48] or a Klystron [49], with efficient DNP occurring at 10 s of watts. A solid state source, with output of up to 250 mW, enables nearly comparable DNP efficiencies at 400 MHz and a broader range of microwave frequencies [22,23]. MAS-DNP NMR instruments operating at ^1^H frequencies up to 900 MHz [16,50,51] are now in operation with MAS probes using rotors of 3.2–1.3 mm outer diameter and reaching spinning speeds of 12–40 kHz. Shown in Figure 1B is a 600 MHz MAS-DNP NMR instrument installed at the National High Magnetic Field Laboratory which uses a quasi-optic setup to control microwave irradiation [20].

#### 2.1.2. Stable Radical Polarizing Agents

Organic biradicals in the form of bis-nitroxides have seen the most success as polarizing agents (PAs) and are the most commonly used source of unpaired electrons in biological MAS-DNP NMR applications. Initial biradical characterizations indicated the importance of sample geometry for efficient DNP, with fixed orthogonal g-tensor orientations and strong dipolar coupled electrons providing maximal enhancements [52,53,54], even if suboptimal [55]. In an effort to make these biradicals compatible with biological systems, chemical modifications to increase water solubility led to the development of the first water soluble biradical, TOTAPOL [17]. A further focus on improving the electron-electron dipolar coupling and electron relaxation properties (T_1e_ and T_2e_) led to the development of AMUPol, which exhibited ~3–3.5-times greater enhancements compared to those achieved with TOTAPOL at moderate magnetic fields [18,56,57,58]. To date, TOTAPOL and AMUPol are the only commercially available biradical PAs for biological systems (Figure 2A). At higher magnetic field strengths (e.g., 14.1–23.5 T), required for high-resolution biomolecular ssNMR spectroscopy, the CE rotor-event becomes less efficient and DNP build up times get longer for a given biradical. There are also lower achievable DNP enhancements for bis-nitroxides at high magnetic fields in part due to the larger span of electron resonance frequencies reducing the effectiveness of each rotor-event, in particular the MW event. This is not observed for hetero biradicals, i.e., containing two radicals with different g-tensors, such as TEMTriPol which perform the best at high field [14,59]. Nonetheless, bis-nitroxides are still very efficient at 14.1 T and DNP enhancements > 100 can be realized for model systems using AMUPol or AsymPolPOK. AsymPolPOK, in comparison to AMUPol, has a stronger electron-electron interaction (both dipolar and exchange) as compared to AMUPol. In addition, it is a charged biradical due to the presence of phosphate groups that may affect the affinity of the biradical with regard to the membrane.

#### 2.1.3. Gains Demonstrated

The accepted mechanism of polarization transfer throughout an NMR sample is that under microwave irradiation a biradical polarizes nearby nuclear spins, which are unobservable in the NMR spectrum due to broadening induced by the biradical (also referred as bleached/quenched nuclei [39,60,61]. This polarization then equilibrates throughout the nuclear spin bath via spin diffusion (Figure 2B) [62]. The time it takes for nuclear spin polarization to reach a steady state is known as the DNP buildup time. The polarization buildup time at an observed nucleus is dependent on the relaxation properties and concentration of the biradical, the coupling strength of the biradical, nuclear spin diffusion rates, and the average distance between biradicals and a particular nuclear spin moiety [26]. This is most easily visualized in comparing the polarization buildup times of materials surfaces relative to their interiors when the biradical is in a surrounding solvent [60,61,63,64]. A similar phenomenon occurs when using water-soluble biradicals to polarize lipid membranes—the fatty acyl chains will polarize at a slower rate than the glycerol head groups.65 For nuclear spins further away from the PA, spin relaxation in competition with the buildup of DNP polarization can lead to measurably lower achievable DNP enhancements. In lipid membranes, this can lead to enhanced polarization of the surrounding solvent and the lipid headgroup region relative to the membrane interior [65,66]. Proton polarization enhancements can be improved by diluting the nuclear spin bath through deuteration to minimize proton dipolar induced relaxation [67].

### 2.2. Sample Preparation Considerations

Traditional DNP sample preparation for biological systems involves dissolving or suspending the biomolecular assembly of interest in a DNP matrix, which generally consists of an organic biradical dissolved in a buffered water/glassing agent mixture. The glassing agent (typically 30–60% (*v*/*v*) glycerol) is used in an effort to preserve a uniform distribution of PA in the aqueous buffer when the sample is slowly frozen in the MAS-DNP NMR probe. Upon cooling a sample to the cryogenic temperatures where DNP is performed (~100 K), the DNP matrix forms an amorphous glass. Glassing agents commonly used in MAS-DNP NMR experiments are glycerol-d_8_ and DMSO-d_6_, owing to their miscibility in water, historic use as cryoprotectants, and the favorable biradical electron relaxation properties observed in these mixtures below the glass transition temperature (T_g_) [68,69,70]. The most commonly used DNP matrix, known as “DNP Juice”, consists of 60:30:10 (*v*/*v*) glycerol-d_8_/D_2_O/H_2_O. The high glycerol content (~23 mol%) ensures glass formation at 100 K (T_g_ ~ 170 K), cryoprotection, and a uniform distribution of biradicals [68,71,72]. Deuteration enables longer ^1^H spin relaxation times, which promotes efficient magnetization transfer within the proton network and results in larger DNP enhancements. Proton polarization is then transferred to low-gamma spins, such as ^13^C and ^15^N, via cross polarization [73]. Large enhancements have been reported for model compounds in DNP Juice [43,53] as well as many protein samples [74] and thus it has been adopted as the standard in DNP sample preparation.

#### 2.2.1. Biradical Polarizing Agents and Lipid Membranes

Membrane peptides and proteins pose unique challenges for MAS-DNP, requiring a DNP matrix that (1) preserves the native lipid membrane environment, (2) efficiently and uniformly polarizes proteins deeply embedded in the lipid bilayer, and (3) allows standard incorporation of known PA concentrations into liposome suspensions. For lipid membranes the high glycerol content of “DNP juice” can be problematic. For example, at glycerol molar concentrations of 5–50% in aqueous DPPC multilamellar preparations, glycerol partitions into the lipid glycerol headgroup region and alters lipid spacing and interactions [75]. At ~50 mol%, the glycerol induces lipid interdigitation, which affects lipid head group mobility and membrane packing. Glycerol also affects the surface and bulk water hydrogen bonding networks, which ultimately impacts lipid membrane hydation [76]. To circumvent this, “matrix-free” sample preparation strategies have been tested for membrane systems. These include using chemically modified PAs that allow for direct incorporation into lipid bilayers or that are attached to membrane peptides or proteins [43,65,77,78,79,80,81,82,83,84]. In addition to incorporating deuterated solvents, lipid deuteration has also been shown to increase DNP enhancement [65,66]. Despite these advances, sample uniformity, in particular, PA distribution and PA concentrations in membrane samples are difficult to systematically control for in both DNP matrix and matrix-free approaches. These effects can lead to low DNP enhancements, irreproducible DNP performances between similar samples or experimental runs of the same sample, and denaturation of protein structure [65]. Below, we show how using a rational, systematic approach to sample conditions one can achieve maximal and reproducible DNP signal enhancements using AMUPol while preserving a native-like lipid membrane environment.

#### 2.2.2. Freezing of Samples

An alternative approach to preserving PA distributions without the addition of a glassing agent is to rapidly freeze the sample before placing it into the rotor [85,86,87]. This technique is less widely adopted due to the difficulty of packing the resulting frozen particles into a rotor while keeping them frozen as well as the larger volume of water used to suspend the sample causing further dilution of the protein of interest. Another consideration in freezing of samples is minimizing the amount of oxygen, a paramagnetic relaxation agent. This can be accomplished to some extent via degassing of buffers prior to their use, but residual oxygen, particularly within the lipid bilayer, may require multiple freeze-thaw cycles during sample preparation and potentially even after the sample is packed into a rotor [88,89]. Once a sample is in the probe, cooled, and spinning, another consideration is what temperature to use for experiments. The lowest temperature possible is dictated by the cryogenic characteristics of the instrumentation as well as the gas used for MAS. For example, the speed of sound of nitrogen gas decreases substantially as one nears its freezing temperature of 77 K so spinning rates in the 90–100 K range are typically two-thirds of what one can accomplish for the same size sample at ambient temperature [40,43]. Nonetheless, DNP enhancements increase with lower temperatures due to spin relaxation times increasing [40,41,43,49]. An additional factor to consider for biological samples is the glassing temperature of the sample; this dependent on both the glassing temperature of the DNP matrix as well as the glassing temperature of proteins and lipid bilayers [90,91]. While literature values exist for conventional glassing mixtures, proteins, and lipids, the true phase transitions within an individual sample can easily be assessed by observing the proton spectrum while cooling the sample. Greater enhancements observed in proteins below 105 K are in part due to freezing of amino acid sidechain motions [92,93]. However, the heterogeneity of the sidechains when frozen can be detrimental to the achievable spectral resolution [94,95]. Thus, particular applications of DNP must consider the tradeoffs between sensitivity and resolution.

#### 2.2.3. Glassing Agents and Buffer Optimization

Sample preparation strategies aimed at preserving the integrity of the lipid membrane environment and membrane protein structure under DNP conditions require further consideration of the glassing agent and buffer. In cellular biology research, 10% (*v*/*v*) DMSO, equivalent to a mole fraction, X, of ~0.03, is routinely added to cell cultures and cellular tissue for storage at cryogenic temperatures. This is sufficient to preserve cellular membrane integrity and >75% cell viability when used in combination with regulated freezing and thawing rates [96,97]. Trehalose can also be used as a cryoprotectant, but is not as robust to freeze/thaw cycles as DMSO, preserving ~50% cell viability. A combination of DMSO and trehalose can preserve >90% cell viability when used in combination with regulated freezing/thawing. Additionally, for 10% (*v*/*v*) DMSO, changes in membrane packing and surface dehydration are not observed [98]. We have found that 10% (*v*/*v*) DMSO provides sufficient cryoprotection and preserves PA distribution for optimal MAS-DNP NMR measurements using multilamellar lipid vesicle (MLV) preparations of membrane active peptides, described below, while 30–60% (*v*/*v*) glycerol leads to alteration of KL_4_ peptide structure [65]. We note that high concentrations of glycerol may not significantly affect structures of larger, more robust transmembrane proteins. However, for amphipathic membrane peptides or regions of transmembrane proteins that reside at the membrane interface, disruption of lipid packing and membrane hydration may lead to non-native conformations or membrane partitioning. Additionally, unlike 60% glycerol, DMSO allows for uniform sample density which promotes ease of membrane sample handling and transfer into a MAS rotor via centrifugation.

In addition to using a cryoprotectant to preserve the PA distribution and membrane integrity, freezing of biologic samples requires consideration of the effect of freezing on pH. Phosphate buffer, a buffer commonly used in biochemistry and molecular biology experiments, is particularly sensitive to both temperature as well as freezing, exhibiting a change of ~2–3 pH units on freezing even when rapidly cooled (5000–20,000 °C/s) [99]. Organic acids, such as HEPES, MES, and NaAC, show much less variability in pH with temperature and freezing. Combining these buffers in a “universal” buffer (UB; 20 mM HEPES, 20 mM MES, 20 mM NaAc) enables facile buffering over a pH range of 2–8 and preservation of pH during freezing [100].

Traditional membrane peptide sample preparations for solid-state NMR spectroscopy follow a standard scheme of mixing peptide and lipids followed by sample hydration. First, lipids and protein are co-dissolved in organic solvents and dried under N_2_ gas. Residual organic solvent can be removed by resuspending the lipid/protein film in cyclohexane, freezing, and lyophilization. This produces a dry, white, lipid/protein powder that is subsequently resuspended in buffer and subjected to 10–15 freeze/thaw cycles to produce MLVs. Notably, the sample needs to be heated above the melting temperature of the lipid mixture during each freeze/thaw cycle to ensure homogenous, well-hydrated MLVs. Following the freeze/thaw cycles, proteoliposomes are pelleted via ultracentrifugation and the excess water layer is discarded. Removal of residual water from proteoliposomes, via N_2_ gas, lyophilization, or using a desiccator with a controlled humidity level, enables more membrane protein to be packed into the NMR rotor, thus increasing sensitivity. These water-removal methods typically rely on monitoring mass to estimate hydration levels and sample equilibration (if regulating via controlled humidity). For MAS-DNP NMR samples, PA dissolved in glassing matrix is often added to the wet proteoliposome pellet prior to removal of residual water [101]. However, this makes regulation of the PA concentration difficult and can potentially lead to deleterious PA aggregation. An alternate approach is to add the PA and glassing matrix directly to the sample at the hydration step so that the concentrations of PA and cryoprotectant are known (Figure 3). Subsequent freeze/thaw cycles then also evenly distribute the PA and cryoprotectant in the MLV suspension before using centrifugation to transfer the sample into an NMR rotor. Upon centrifugation into the NMR rotor, minimal phase separation between the lipids and aqueous buffer may occur. We do not remove excess buffer in the NMR rotor in order to preserve the fixed concentration of PA. This method, albeit slightly reducing the amount of MLVs packed into the fixed volume NMR rotor, maintains a known concentration of PA and promotes a uniform PA distribution and optimal DNP enhancements. The gain in sensitivity afforded by optimal DNP enhancements overcomes the sensitivity loss due to reduced sample concentration. This approach also ensures the maintenance of fully hydrated proteoliposomes as well as pH.

Larger membrane proteins are typically heterologously expressed, solubilized in detergent, and purified before reconstitution in lipid membranes for ssNMR experiments. We note that the same approach for adding PA and glassing matrix can be used for theses preparations. Namely, the reconstituted proteoliposomes in buffer can be pelleted via ultracentrifugation and resuspended in a minimal amount of buffer containing PA and glassing agent. Distribution of the PA and glassing agent in MLVs can be accomplished by freeze/thaw cycle followed by pelleting the sample into a rotor for MAS-DNP NMR experiments.

### 2.3. Evaluation of Samples and DNP Optimization

Biradical concentration and distribution, DNP buildup times, DNP enhancements, and nuclear spin relaxation rates should be evaluated prior to performing traditional ssNMR experiments under MAS-DNP conditions. This allows for rapidly assessing sample quality; making any needed adjustments in PA concentration, buffer, and/or cryoprotectant; maximizing sensitivity gains; and determining feasibility of multidimensional NMR experiments.

#### 2.3.1. Measurements of Biradical Concentration and Distribution

Continuous wave (CW) EPR X-band (9.5 GHz) measurements can validate the concentration and quality of PA in a sample prior to performing DNP experiments. Many benchtop X-band resonators are compatible with standard NMR rotors, enabling direct characterization of a DNP sample just prior to placing it in the MAS-DNP NMR probe. EPR spectra should be collected at both room temperature and ~100 K to assess PA distribution upon cooling to cryogenic temperatures. Well resolved splittings in both RT and 100 K spectra are indicative of a well-dispersed and homogenous PA distribution. At 100 K, EPR spectral line shapes should be broadened due to decreased molecular motion, but the EPR spectrum remains well-resolved if the PA is well distributed. The hallmark of PA aggregation is a complete loss of spectral resolution and broad, featureless EPR spectral line shapes. Biradical breakdown, resulting from heat or incompatible pH, can be detected at room temperature as either a different EPR spectrum or a loss of signal. PA concentration can be evaluated by analyzing doubly integrated EPR spectra or spin counting.

#### 2.3.2. Evaluating DNP Enhancement

A DNP buildup (T_B_) experiment measures the rate of ^1^H polarization enhancement and determines the optimal recycle delay time for subsequent NMR experiments. Buildup times can be measured using the pulse sequence shown in Figure 4. Here, microwaves are on during the entirety of the experiment, indicated as continuous wave (CW) irradiation. An initial train of ^1^H 90-degree pulses removes residual transverse ^1^H magnetization from sample equilibration in the magnet. Next, during a variable delay period, the polarization from unpaired electrons is transferred to nearby protons via the CE and rapidly diffuses throughout the proton bath. Finally, proton magnetization is transferred to low gamma nuclei via cross-polarization for detection. Spectral intensities, taken as a function of variable delay time, are used to generate experimental build-up curves that can be fit to the exponential function, I(t) = I(0) × (1 − exp(−t/T_B_)). The recycle delay time for subsequent NMR experiments is typically set to 1.26 × T_B_ for the resonance of interest with the longest build-up time to afford a compromise between maximum sensitivity and sensitivity per unit time. We note the pulse sequence is not drawn to scale as the presaturation, cross polarization, and acquisition are on the order of msec while polarization buildup is on the order of seconds.

#### 2.3.3. Maximizing DNP Enhancements

DNP enhancements can be quickly estimated by comparing NMR spectra collected with MW on (CW) versus MW off. However, this comparison does not control for sample bleaching/depolarization by the biradical or loss of polarization under MAS [102]. A simple ratio will in many cases over-estimate the improvements in SNR from MAS-DNP if the biradical concentration is too high or the PA depolarizes under MAS. A more robust analysis is to compare spectra for a sample containing PA to an identically prepared sample that does not contain PA. Due to the cost of making membrane protein samples and the variability in sample packing efficiencies, this degree of rigor is rarely warranted. A compromise is to use a PA which is well-characterized using model samples at the MAS rates and magnetic field that will be used. For example, AMUPol has been demonstrated to have minimal depolarization at NMR magnetic fields and moderate spinning speeds 39. The concentration of PA to utilize can be evaluated on the basis of measured DNP buildup times and nuclear T_2_ relaxation rates. Increasing PA to an optimal concentration will shorten DNP buildup times and increase true DNP polarization enhancements without dramatically shortening nuclear spin relaxation times. Beyond this optimum, nuclear spin relaxation rates will increase, sample bleaching will occur, and true polarization enhancement will decrease. The optimum point for model samples containing AMUPol is on the order of 5–10 mM for 400–600 MHz. For biomembrane samples, in which the PA may partition into the lipid membrane, or for proteins that bind PA, the optimum sample concentration may be substantially lower [29,103]. This can quickly be assessed by measuring nuclear T_2_ times for a particular sample and comparing them to model or control samples.

## 3. DNP-Enhanced NMR Characterization of the Membrane Active Peptide KL_4_: A Case Study

As an example of how MAS-DNP NMR can transform ssNMR structural biology of membrane proteins, we assessed the improvements in SNR which would be gained using DNP while maintaining ssNMR characteristics indicating a well-structured peptide that can be rigorously characterized by multidimensional NMR experiments for the membrane active peptide KL_4_. KL_4_ is a 21-amino acid peptide developed as a mimetic of pulmonary surfactant protein B. It exhibits adaptive helicity and membrane partitioning as a function of lipid composition and pH [104,105,106,107,108]. Most notably, we identify conditions where the peptide structure, as assessed by chemical shift, is insensitive to the presence of PAs. This enables us to structurally characterize the KL_4_ peptide in its native membrane environment under conditions replicating clinical peptide/lipid formulations for respiratory distress syndromes.

### 3.1. Sample Preparation

KL_4_ (KLLLLKLLLLKLLLLKLLLLK) was synthesized via Fmoc solid phase peptide synthesis (CEM Corporation, Charlotte, NC, USA). Isotopically enriched leucine, [1-^13^C] L-Leucine (CIL, Tewksbury, MA, USA) was purchased and modified for Fmoc solid phase peptide synthesis [109]. Crude peptide was purified via RP-HPLC, and purity was confirmed via MALDI mass spectrometry with [M+H^+^] = 2469.83. Pure KL_4_ was dissolved in methanol to yield a 1 mM monomer peptide solution, with concentration and purity confirmed via analytical RP-HPLC. DPPC, POPG, and DPPC-d_62_ were purchased as chloroform solutions from Avanti Polar Lipids (Alabaster, AL, USA) and quantified by phosphate analysis (Bioassay Systems, Hayward, CA, USA). Cholesterol was also purchased from Avanti Polar Lipids as a powder and dissolved in chloroform at 25 mg/mL. Following lipid quantitation, stock lipid mixtures were prepared by combining individual lipid chloroform solutions. KL_4_ in methanol was added to lipid chloroform mixtures to achieve 2.0 mol% peptide relative to total lipids. Lipid/peptide samples were dried under N_2_ gas, resuspended in warm cyclohexane (>45 °C), frozen in liquid nitrogen, and lyophilized overnight to remove residual solvent. Lyophilized lipid/peptide powders (~20 mg) were first hydrated in 1.0 mL of 10X Universal Buffer (UB), pH 7.4, made with 90/10 (*v*/*v*) 9:1 D_2_O:H_2_O, and equilibrated with 15 freeze/thaw cycles. For AMUPol-containing samples, liposomes were subsequently ultracentrifuged to obtain phase separated proteoliposomes and buffer layers. The buffer was carefully removed with a syringe, and 50 μL of 10 mM AMUPol in 10× deuterated UB containing 10% (*v*/*v*) DMSO-d_6_ was added to the liposome pellet and an additional 15 freeze/thaw cycles were performed. The lipid and DNP matrix suspensions were then vortexed to create a uniform suspension and pelleted into 3.2 mm sapphire NMR rotors and sealed with silicone plugs. The final NMR samples consisted of ~10 mg of proteoliposomes in a 20–25 μL volume.

### 3.2. Evaluation of Samples by X-Band EPR Spectroscopy

CW EPR spectra were collected on a Bruker X-Band EMX Nano benchtop spectrometer (Bruker Biospin, Billerica, MA, USA). DNP sample rotors were placed in 5 mm quartz sample tubes and placed into the cavity with its associated adaptor. Spectra were collected at room temperature (298 K) and at 100 K by using cold nitrogen gas to cool the cavity. Spectra recorded at both RT and 100 K allow assessment of PA concentration and distribution at cryogenic temperatures prior to performing DNP experiments (Figure 5). In our samples, we observed no evidence of PA aggregation as spectral features for AMUPol are resolved at both temperatures. At room temperature, the spectra are typical of AMUPol in an aqueous medium [56]. The loss of spectral resolution at 100 K is the result of decreased molecular motion at 100 K. Spectra recorded for a suite of DNP samples containing PA allow for assessment of sample quality and sample-to-sample heterogeneity. Figure 5 shows the CW X-band EPR spectra for three membrane samples prepared in parallel with identical additions of PA in buffer. Samples 2 and 3 contain 2 mol% KL_4_ whereas sample 1 does not contain peptide. For samples containing peptide, we observed a slightly different AMUPol spectrum than for the sample that did not contain peptide. One potential explanation for this difference is that the PEG-modification in AMUPol leads to partial biradical partitioning at the lipid interface, which would decrease AMUPol motion, and that the peptide mediates this partitioning.

### 3.3. Evaluation of DNP Enhancement

MAS-DNP NMR spectra were collected at 600 MHz (14.1 T) using a Avance III DNP system (Bruker Biospin) with a Bruker 3.2 mm, ^1^H/^13^C/^15^N MAS-DNP probe cooled to ~95 K. Microwaves were delivered with a gyrotron source operating at 395 GHz and output optimized for ~12 W at the probe base. Microwaves on/off spectra were collected with a MAS frequency of 10 kHz and a ^13^C CP-echo pulse sequence with a ^1^H CP ramp from 40 to 70 kHz, a ^13^C RF field of 50 kHz, 100 kHz SPINAL-64 ^1^H decoupling and a recycle delay of 5 s. An initial comparison of spectra for MW on vs. MW off, shown in Figure 6, indicates DNP enhancements of ~20 fold.

Signal intensities in microwave on and microwave off spectra, normalized for the number of scans, were used to calculate DNP enhancements ($\varepsilon_{on/off})$ shown in Table 1. For most resonances, these ratios reflect absolute sensitivity gains due to the absence of paramagnetic bleaching at 10 mM PA concentration, as determined by comparison to a control sample made without biradical (Figure 7). We note a minimal level of sample bleaching for lipid glycerol resonances (35–75 ppm) and the buffer resonance at 181 ppm. Notably, there is a considerable difference in enhancements between samples. In particular, the KL_4_ containing samples show significantly less enhancement of the lipid moieties relative to the sample made without peptide. This again suggests the peptide affects partitioning of the PA within the samples. Nonetheless, enhancements of 8–58 are seen across the different samples and resonances, showing a substantial increase in sensitivity from DNP. Between the two peptide-containing samples we note a difference in enhancements even though they were prepared in parallel using the same level of PA. Differences in overall PA levels were not detected by EPR or as sample bleaching, suggesting the primary cause of differences in enhancement is the distribution of the PA within the MLVs.

In comparing the enhancements for the lipid functional groups, the glycerol region consistently showed the greatest enhancement, consistent with the AMUPol being excluded from the membrane interior. The glycerol region also showed greater enhancement than the buffer in the sample made without peptide, suggesting partitioning of AMUPol at the membrane interface. In the peptide containing samples, greater relative polarization of the buffer was observed suggesting again that the peptide diminishes AMUPol partitioning at the membrane interface. These results are consistent with a previous study where paramagnetic relaxation (PRE) effects observed at ambient temperature indicate preferential localization of AMUPol and TOTAPOL biradicals in the lipid membrane [101]. The glycerol region also showed greater enhancement than the buffer in the sample made without peptide, suggesting partitioning of AMUPol at the membrane interface.

To gain more insight into the underlying causes of differences in polarization between samples, we also compared DNP build up times (Table 2). For simplicity, if one assumes a long nuclear relaxation time, T_1n_, T_B_ reports on the efficiency of polarization transfer from electrons to nearby protons and subsequent diffusion. It is an indirect probe of biradical distribution within a sample. At 10 mM PA concentration and 600 MHz/395 GHz DNP conditions, typical T_B_ times of ~3–5 s are expected for samples with uniformly-dispersed PA based on measurements made with model samples [110]. This was similarly observed for the buffer and the glycerol headgroups in the sample containing MLVs without peptide, thus indicating a uniformly dispersed biradical distribution, or polarization source, within the buffer and at the lipid interface. A clear difference is seen in polarization build up times between the buffer and glycerol moieties compared to the membrane interior, suggesting more extended spin diffusion is needed for polarization to reach the hydrophobic lipid interior. While the faster buildup times for the buffer and glycerol headgroups suggest close proximity to the PA, the differences in enhancements and minor sample bleaching observed for these resonances suggest subtle variations in local PA concentration may lead to depolarization or quenching of the lipid glycerol region and buffer relative to the rest of the sample and an overestimation of DNP enhancements for these moieties. For the peptide-containing sample which exhibited the lowest enhancement, the polarization buildup times are substantially faster for all positions. This suggests a higher local concentration of the PA, or poor dispersion of the PA throughout the MLVs, leading to an ineffective CE mechanism, due to deleterious spin relaxation, and inhibition of polarization transfer throughout the sample. For the peptide-containing sample exhibiting higher enhancements, polarization buildup times and enhancements for the buffer are similar to the sample containing only lipids suggesting equally ideal distribution of the PA in the aqueous layer. Interestingly, the polarization buildup times are only slightly longer for the membrane interior relative to the buffer in contrast to the substantially lower enhancements (13–14 in contrast to 22–36). Additionally, the glycerol region shows less enhancement in this sample compared to the sample containing only lipids. These lower enhancements combined with shorter T_B_ times suggest that the peptide, which is fully protonated and contains many methyl groups, induces T_1n_ relaxation changes which are observed as reduced enhancements at the lipid moieties. This enhanced relaxation plays against the DNP process and prevents observed resonances from reaching high polarization levels, thus leading to lower $\epsilon_{\mathrm{on}/\mathrm{off}}$.

### 3.4. Evaluation of Nuclear Spin Coherence Times

In order to assess the viability of DNP samples for more complex NMR experiments requiring longer transverse relaxation times, we assessed the impact of PA on ^13^C T_2_ relaxation times in the samples with well-dispersed PA (Table 3). A control sample made without PA was used to benchmark ^13^C T_2_ relaxation times in these samples at cryogenic temperatures. T_2_ was measured using a rotor-synchronized Hahn-echo sequence and data were fit to a mono-exponential decay function. The addition of PA clearly lowers the ^13^C T_2_ relaxation times, with the largest degradation seen for the buffer resonances. This is unsurprising given the localization of the PA in the aqueous phase. Somewhat surprisingly, a similar degradation in the T_2_ relaxation times were not observed for the lipid glycerol moieties, suggesting proton dipolar couplings may still dominate transverse relaxation for protonated spins. Equally heartening, T_2_ relaxation times for the other moieties, while affected by PA, are quite long and on the order of T_2_ relaxation times we typically observe at ambient temperatures—the drop in T_2_ relaxation times as a consequence of adding PA is essentially cancelled by lowering temperature by ~200 K. This suggests a bright future for standard biomolecular ssNMR pulse sequences applied under MAS-DNP conditions.

### 3.5. NMR Evaluation of Sample Integrity

Previously we have observed that the use of DNP juice leads to destabilization of KL_4_ peptide structure.65 This can be evaluated by comparing sample intensities at 176 ppm, typical for KL_4_ in DPPC rich bilayers at pH 7.4 as observed by ssNMR experiments, vs. 172 ppm, a less structured form of KL_4_ observed in POPC rich bilayers and at lower pH. Under MAS-DNP conditions, this comparison is more challenging due to overlap with the lipid resonance at ~173 ppm, but nonetheless a clear resonance can be seen at 176 ppm. Spectral deconvolution of the resonances at 170–180 ppm in peptide-containing samples and comparison to the sample that did not contain peptide suggest that the peptide is exclusively in a helical conformation at 176 ppm within the resolution limitations of the samples. The peptide resonances have FWHM line widths of ~2.5–3.0 ppm, a ~1.5 ppm increases in line width compared to spectra we have collected via conventional NMR at 238 K for samples prepared with no polarizing agent.

## 4. Conclusions and Future Prospects for DNP of Membrane Proteins

Comparison of DNP enhancements for MLV sample preparations containing the surfactant peptide KL_4_, which is sensitive to both lipid partitioning and pH, allowed assessment of whether the utilization of water-soluble PA, namely 10 mM AMUPol, could be routinely used to polarize membrane protein preparations and give reproducible DNP enhancements. In an effort to minimize inconsistencies in sample handling, all samples were prepared together and utilized the same stock solutions of lipid, peptide, and polarizing agent. Based on X-band EPR measurements and assessments of DNP enhancements, PA can be routinely incorporated into MLVs using a simple freeze/thaw approach for dispersing the PA. The addition of 10% DMSO provides sufficient cryoprotection to maintain dispersion of the PA in the aqueous phase while preserving membrane and peptide structural integrity. Nonetheless, significant differences in enhancements between samples are observed due to (a) less efficient DNP polarization due to increased longitudinal nuclear relaxation in peptide containing samples, and (b) sample-to-sample variability in equilibrating PA throughout the MLVs. The former source of variability may be overcome with deuteration of peptide residues at methyl positions and the latter can likely be overcome by increasing the number of freeze/thaw cycles standardly used while the former source of variation will be dependent on the lipids and the membrane proteins or peptides in a given sample. The use of 10% DMSO and freeze/thaw equilibration of PA within MLVs is likely to produce superior reproducibility of PA dispersion and sample integrity in general for membrane protein samples compared to previously employed sample preparations utilizing “DNP juice” with 60% glycerol as the DNP matrix and an “add on top” approach where PA dissolved in DNP matrix is added to a membrane protein sample with minimal hydration. This is due to better matching of viscosity between the MLVs and the DNP matrix made with 10% DMSO, less dehydration at the membrane interface by DMSO, and better control of PA concentration within the samples by using a slight excess of matrix with a known PA concentration. Importantly, organic acid-based buffers, which exhibit minimal pH changes with temperature and freezing, should be utilized in sample preparations.

Evaluation of DNP enhancements and DNP buildup times can quickly provide valuable assessments of PA distribution within samples and enable optimization of sample conditions for maximal DNP gains. Variability in DNP enhancements and buildup times throughout lipid membranes due to partitioning of water-soluble PA within the aqueous phase or at the membrane interface can provide valuable guideposts for evaluating membrane proteins samples with limited prior information regarding the localization of protein domains relative to the membrane environment.

Under DNP conditions, ^13^C spin T_2_ relaxation times are slightly shortened by addition of PA but spin coherence lifetimes are comparable to those measured by ssNMR at ambient temperatures. This suggests most ssNMR pulse sequences can be employed in MAS-DNP NMR experiments without any detrimental loss of signal in multidimensional NMR experiments. We note that inhomogeneous broadening of resonances for the samples we studied was a factor of 1.5–2-times worse than for similar samples frozen and studied above the protein glass transition via ssNMR spectroscopy.

The added sensitivity of DNP, combined with robust sample preparation methodologies for membrane proteins, will likely lead to routine use of MAS-DNP NMR approaches to garner one to two orders of magnitude improvements in sensitivity relative to conventional ssNMR. This will fundamentally alter the application of NMR spectroscopy in membrane protein structural biology. In particular, with improved PAs (TEMTriPol-I, AsymPolPOK, bcTol-M), and very low temperature MAS-DNP (He recirculating systems), the improved sensitivity of NMR will open new possibilities. Such technology will indeed extend biomolecular NMR methodologies to nuclei which are typically not observed by ssNMR due to sensitivity limitations, such as ^17^O and ^43^Ca. This will enable fundamental studies of chemical mechanisms within membrane proteins.

## Figures and Tables

**Figure 1 biomolecules-10-01246-f001:**
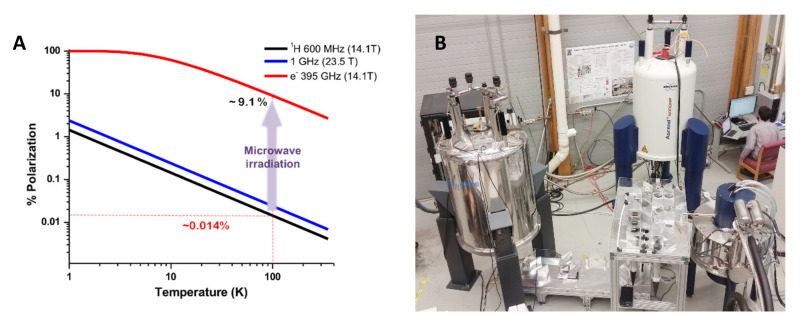
(**A**): Electron and nuclear polarization as a function of temperature and magnetic field. The polarization of an electron (red solid line) and ^1^H (black line) at 100 K and 14.1 T are highlighted (red dashed line). The ^1^H polarization gain with moving to a higher magnetic field (blue line) is shown for comparison. The general scheme of dynamic nuclear polarization (DNP) is to irradiate at the resonance of an unpaired electron spin (395 GHz at 14.1 T), resulting in the transfer of polarization to nuclear spins, thereby giving DNP enhancement. (**B**): 600 MHz DNP installation at the National High Magnetic Field Laboratory. The magic angle spinning (MAS)-DNP solid-state nuclear magnetic resonance (ssNMR) system is pictured in the upper right corner along with the gyrotron in the lower right corner. The gyrotron supplies high power microwave irradiation to two magnets via the quasioptic tables in the center of the photograph. A complete description of this instrumentation has been published [20].

**Figure 2 biomolecules-10-01246-f002:**
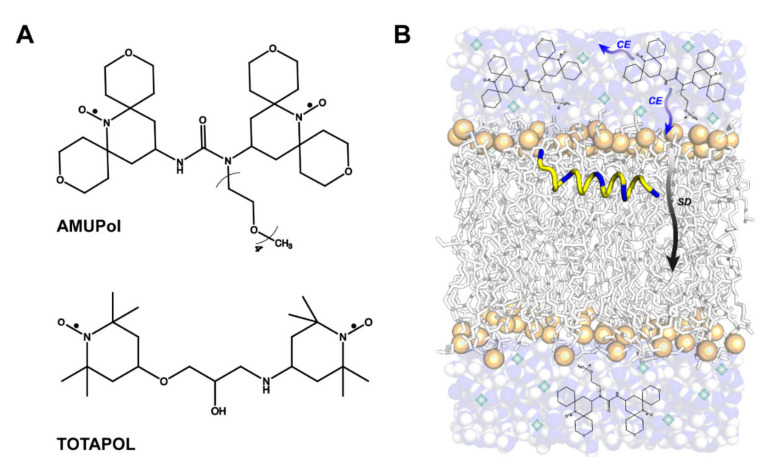
(**A**) Commercially available water-soluble biradicals for DNP. (**B**) Likely polarization transfer pathways via cross-effect DNP (CE) and spin diffusion (SD) in biomembrane samples incorporating AMUPol. Green diamonds indicate the presence of a glassing agent to maintain AMUPol dispersion during sample freezing.

**Figure 3 biomolecules-10-01246-f003:**
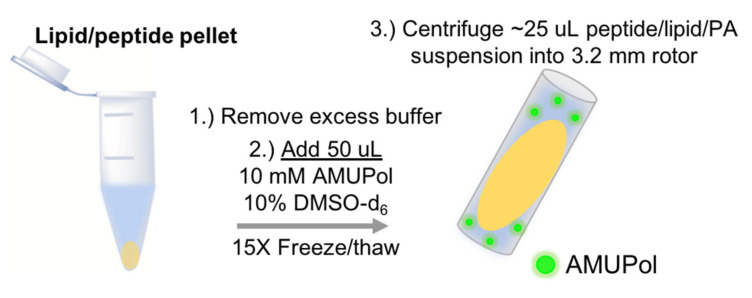
Cartoon schematic of proteoliposome sample preparation for MAS-DNP. The lipid/peptide multilamellar lipid vesicle (MLVs) are produced following traditional membrane peptide or protein sample preparations for ssNMR. Here, the lipid pellet is first hydrated in excess buffer and subjected to 15X freeze/thaw cycles to form homogenous MLVs and pelleted by ultracentrifugation. Following this, steps 1–3 illustrate addition of polarizing agents (PA), and additional freeze/thaw cycles to distribute the PA and glassing agent prior to sample transfer into the NMR rotor.

**Figure 4 biomolecules-10-01246-f004:**
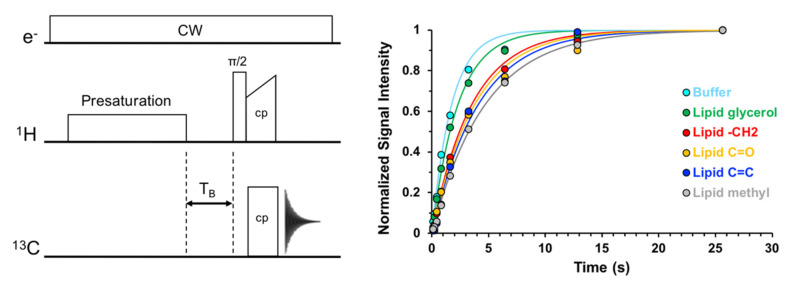
Pulse sequence for DNP buildup time (T_B_) measurements and example data for a sample containing lipid bilayers and 10 mM AMUPol as the polarizing agent.

**Figure 5 biomolecules-10-01246-f005:**
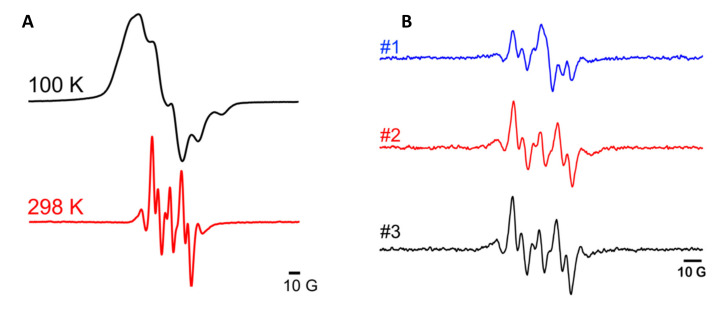
(**A**): Continuous wave (CW) EPR X-band (9.5 GHz) spectra of 10 mM AMUPol in a sample containing 5.5/2.7/2/1 DPPC-d_62_/DPPC/POPG/cholesterol liposomes with 2 mol% KL_4_ peptide. EPR spectra shown in red and black were collected at 298 K and 100 K, respectively. (**B**): EPR spectra at 298 K for three liposome preparations containing 10 mM AMUPol prepared in parallel. These EPR spectra provide a qualitative assessment of PA distribution between identically prepared biomembrane samples. Sample one contains DPPC-d_62_/DPPC/POPG/cholesterol liposomes without peptide; samples 2 and 3 contain lipids and 2 mol% KL_4_ peptide.

**Figure 6 biomolecules-10-01246-f006:**
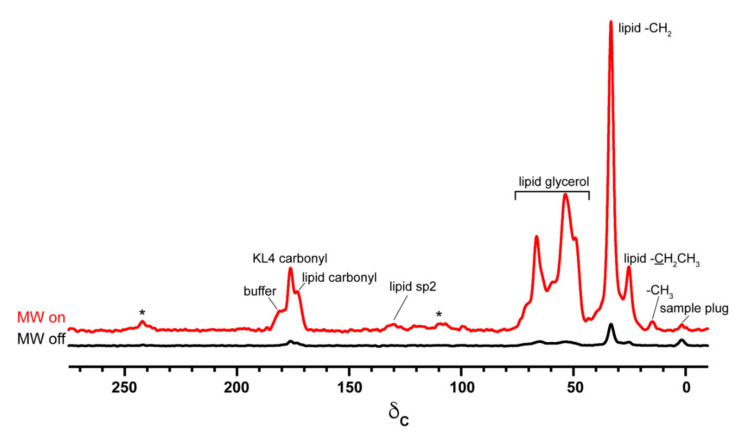
^13^C chemical shift spectra collected with MW off (black) or microwave on (red) for a sample containing 5.5/2.7/2/1 DPPC-d_62_/DPPC/POPG/cholesterol MLVs with 2 mol% K_4_ equilibrated in a buffer containing 10% DMSO-d_6_ and 10 mM AMUPol. Spectra were scaled based on the number of transients averaged. The “MW on” spectrum is an average of 32 transients collected in ~2 min; the “MW off” spectrum is an average of 512 transients collected in ~35 min. MAS sidebands are indicated by an *. Since the DMSO is deuterated, solvent resonances contribute minimally to the spectrum.

**Figure 7 biomolecules-10-01246-f007:**
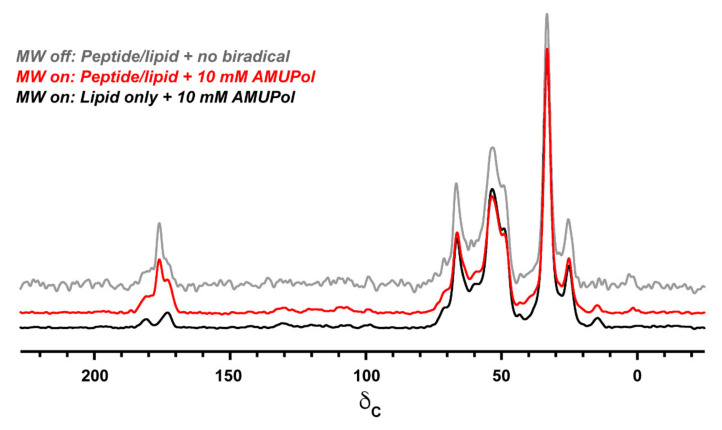
Comparison of ^13^C chemical shift spectra for a DNP-enhanced sample containing 5.5/2.7/2/1 DPPC-d_62_/DPPC/POPG/cholesterol and 2 mol% K_4_ made with buffer containing 10% DMSO and 10 mM AMUPol (red) vs. a DNP spectrum for a control sample made without peptide (black) and an NMR spectrum for a control sample made without polarizing agent (grey). The DNP spectra are an average of 32 transients collected in ~2 min; the control NMR spectrum is an average of 512 transients collected in ~35 min. Spectra are scaled to the lipid -CH_2_ resonance in the lipid only sample and no appreciable differences in resolution were detected between the samples.

**Table 1 biomolecules-10-01246-t001:** DNP enhancements for different samples. All samples were comprised of 5.5/2.7/2/1 DPPC-d_62_/DPPC/POPG/cholesterol lipids reconstituted in UB, pH 7.4, containing 10 mM AMUPol and 10% DMSO. Peptide-containing samples incorporated KL_4_ with a single ^13^C isotope at the carbonyl carbon for specific amino acid positions. Site-specific ^13^C-labelled KL_4_ peptides at residues L12 or L12, L13 were synthesized and peptides were reconstituted at 2 mol% relative to the lipids. Enhancement values are reported with ±10% accuracy due to limited SNR for microwaves off spectra.

Sample	Buffer	Lipid Glycerol	Lipid C=O	Lipid CH2	Lipid C=C	Peptide C=O
KL4/Lipids-1	36	22	13	13	14	13
KL4/Lipids-2	23	15	8	8	N.D.	8
Lipids only	34	58	48	45	34	--

**Table 2 biomolecules-10-01246-t002:** DNP buildup times for different samples. All samples were comprised of 5.5/2.7/2/1 DPPC-d_62_/DPPC/POPG/cholesterol lipids reconstituted in UB, pH 7.4, containing 10 mM AMUPol and 10% DMSO. Peptide-containing samples consisted of KL_4_ with a single ^13^C isotope at the carbonyl position. Site-specific ^13^C labelled KL_4_ peptides at residues L12 or L12,L13 were synthesized and peptides were reconstituted at 2 mol% relative to the lipids. Buildup times are based on fitting data to nine time points and are reported with 95% CI.

Sample	Buffer	Lipid Glycerol	Lipid C=O	Lipid CH2	Lipid C=C	Peptide C=O
KL4/Lipids-1	2.7 ± 0.7 s	2.2 ± 0.2 s	3.5 ± 0.3 s	3.3 ± 0.2 s	4.2 ± 0.7 s	3.6 ± 0.1 s
KL4/Lipids-2	1.4 ± 0.1	1.6 ± 0.1	2.3 ± 0.2	2.2 ± 0.1	2.0 ± 0.6	1.8 ± 0.1
Lipids only	3.2 ± 0.6	3.5 ± 0.4	8.8 ± 0.5	7.7 ± 0.2	8.1 ± 1.2	--

**Table 3 biomolecules-10-01246-t003:** ^13^C T_2_ relaxation times for different samples measured using a rotor-synchronized Hahn-echo sequence. All samples were comprised of 5.5/2.7/2/1 DPPC-d_62_/DPPC/POPG/cholesterol lipids reconstituted in UB, pH 7.4, containing 10 mM AMUPol and 10% DMSO, except the control sample which did not contain AMUPol. The peptide-containing samples, indicated as KL4/Lipids or Control, had KL_4_ labeled with a single ^13^C isotope at the carbonyl position in L12 and peptides were reconstituted at 2 mol% relative to the lipids. Relaxation times for DNP samples are based on fitting data to multiple (>10) time points and are reported with 95% CI. For the control sample lacking biradical, only three time points were collected due to signal limitations.

Sample	Buffer	Lipid Glycerol	Lipid C=O	Lipid CH2	Peptide C=O
KL4/Lipids	15 ± 4 ms	4.9 ± 0.3 ms	33 ± 3 ms	8.0 ± 0.7 ms	37.1 ± 1.5 ms
Lipids only	9.9 ± 0.9	4.3 ± 0.2	35 ± 2	6.9 ± 0.5	--
Control	82 ± 20	3.2 ± 0.9	73 ± 5	11 ± 4	64 ± 18

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
