# Peer review of "Dynamic Nuclear Polarization of Biomembrane Assemblies"

_biomolecules, 2020, doi:10.3390/biom10091246_

Round 1

Reviewer 1 Report

Tran et al. present a detailed discussion of sample preparation methodology for DNP studies of membrane systems, with discussion of the glass forming agent as well as the buffer. While the procedures are not particularly new, the detailed presentation will be very helpful for the DNP community. I recommend publication with only minor changes, suggested below:

  1. In general, I do not find the sonication to be well supported by the data shown. There was variability in the sonicated samples (mentioned line 565). Did the authors also try mechanical mixing of the sample and waiting for biradical to diffuse through membranes?
  2. in the introduction, page 2 lines 56-57, it would help to be more specific about the recent improvements and how much is left to do. What is now the primary limitation to be overcome?
  3. page 2 line 67 mentions 'biologically relevant concentrations' Is that pM, nM, uM?
  4. page 3 line 118 states that polarization difference... [is] not maintained. Is it generated at all (speaking of polarization as a bulk property, not a local one)?
  5. page 4, ref 66, Mei's work is also relevant here (Liao, J.Biomol.NMR 2016) and specifically discusses biradical partitioning. Also page 12 line 470, is this consistent with Mei's work?
  6. page 7 line 257 states that glycerol 'leads to loss of peptide structure'. This statement is far too general, since it refers only to certain peptides, not to peptides in general.
  7. then on line 258 there is a statement about 'uniform viscosity' that does not make sense or needs clarification. Membrane samples are routinely handled with glycerol without problems.
  8. page 7 line 264. 'rapidly cooled' What is rapid? (e.g. in degrees per second)
  9. Figure 3 is a bit unclear. 'add 50 uL' should be comparable to the sample amount in some way without digging in the text. (or state as a percent or fold excess). Is the centrifugation to the rotor also a separation?
  10. page 11 lien 442, references 'bleaching at this PA concentration' . Which concentration?
  11. I find the scaling discussion in fig. 7 to be impossible to follow. If they are signal averaged, why is there any additional scaling, and if there is, it should be from the enhancement, not the number of scans. A legend would help.
  12. The coherence lifetimes part needs to be reworked. Page 14 and table 3 discuss T2 and T2* (full linewidths), but the table 3 gives values in seconds, which can only be buildup times. Overuse of 'coherence lifetimes' rather than specific mention of T2 T2 prime T2 star, T1, T1rho is too imprecise. (This also applies to the statement line 586.) It is unclear if the discussion is about spin echo measurements, or build ups, or what specific measurement was done that has '>10' time points. Lange et al. JMR 2012 and Corzilius et al. JMR 2014 are relevant citations related to the discussion on coherence times.

Typos:

  • page 2 line 66 'proteindilution' make two words
  • page 3 line 119 'alarge' -> two words
  • page 7 line 283 N2(g) -> N2 gas
  • page 11 line 428 MAS speed -> MAS frequency
  • line 545. 'this is comparison' -> this comparison
  • line 599 'lot observed' -> not observed

Reviewer 2 Report

Tran, Mentik-Vigier and Long provide a great technical review on dynamic nuclear polarization (DNP) applied to membrane proteins. They cover various aspects of the topic including the physical phenomenon of DNP, radicals available, sample preparation and NMR aspects of measurement with a nice focus on the specificities of membranes. Lastly, they include a case study on the membrane active peptide KL4. Although I found the review well-structured and well written, it would benefit from a thorough readthrough for typos, missing punctuations, etc.

My only major comment would be that, since the main drawback of using DNP is always the loss of resolution, the authors could provide a discussion or a few tips on the ideal sample preparation and measurement parameters to minor resolution loss. This can easily be coupled with the paragraph on freezing and temperature (L219-243).

Also, I found that a few important references on the topic were missing:

  • Mak-Jurkauskas et al. PNAS 2008 105(3) p.883
  • Bajaj et al. PNAS 2009 106(23) p.9244
  • Van der Cruijsen et al. Chemistry 2015 21(37) p.12971
  • Wylie et al. JBNMR 2015 61(3) p.361

The following are minor comments/ideas for improvements:

  • L48 – here the authors could provide the actual number since it is theoretical
  • L66 – space missing
  • L151 – clarification of what “hetero biradicals” mean in the context of TEMTriPol is required here
  • L153 – the authors should write a short line about AsymPolPOK and how it is different from AMUPol
  • L159 – the first paper in which the spin diffusion barrier effect has been observed is Debelouchina et al. 2010 Phys Chem Chem Phys 12(22) p. 5911 and should be cited
  • L210 – one more reference could be included here: Viennet et al. 2016 Angew Chem 55(36) p.10746
  • L336 – while I agree that the 1.26*TB is theoretically the maximum sensitivity per unit time that can be reached, in my experience using more scans and reduced recycle delays always provides better sensitivity. Have the authors tested this? If not, maybe it can be made more apparent that this is theoretical and might be optimized experimentally for a given system.
  • Line 440 – if the on and off spectra were acquired with different number of scans, the enhancement can be biased by incomplete recycling as the authors nicely pointed out beforehand. Was the TB measured?
  • Line 450 – I find confusing those two peptide-containing samples, they seem to be the same but behave very differently. Is there any difference in the sample preparation protocol at all?
  • Figure 1 – it is impossible to read what’s inside the arrow. The photograph might be easier to understand with arrows pointing at the different instruments rather than “top right”, etc. in the caption.
  • Figure 2 – the chemical structures look somehow blurry, higher resolution would be nicer
  • Figure 5 – the difference between sample on the left and sample 1 is not clear, as is the difference between samples 2 and 3. Captions shall be made clearer.
  • Figure 6 – what are the stars and where is the DMSO?
  • Table 2 – the labeling schemes and ratios for each sample is not clear. Caption should be made clearer. L511 is it “add” or “are”?

Author Response

We thank the reviewer for their careful reading of our manuscript and their helpful critiques. We have answered all critiques in our updated manuscript. Please see the attachment for further details.
